# Lung Radiomics Features Selection for COPD Stage Classification Based on Auto-Metric Graph Neural Network

**DOI:** 10.3390/diagnostics12102274

**Published:** 2022-09-20

**Authors:** Yingjian Yang, Shicong Wang, Nanrong Zeng, Wenxin Duan, Ziran Chen, Yang Liu, Wei Li, Yingwei Guo, Huai Chen, Xian Li, Rongchang Chen, Yan Kang

**Affiliations:** 1College of Medicine and Biological Information Engineering, Northeastern University, Shenyang 110169, China; 2College of Health Science and Environmental Engineering, Shenzhen Technology University, Shenzhen 518118, China; 3School of Applied Technology, Shenzhen University, Shenzhen 518060, China; 4Department of Radiology, the First Affiliated Hospital of Guangzhou Medical University, Guangzhou 510120, China; 5Shenzhen Institute of Respiratory Diseases, Shenzhen People’s Hospital, Shenzhen 518001, China; 6The Second Clinical Medical College, Jinan University, Guangzhou 518001, China; 7The First Affiliated Hospital, Southern University of Science and Technology, Shenzhen 518001, China; 8Engineering Research Centre of Medical Imaging and Intelligent Analysis, Ministry of Education, Shenyang 110169, China

**Keywords:** COPD stage (GOLD), auto-metric graph neural network (AMGNN), multi-classification, lung radiomics features, Lasso algorithm, generalized linear model (GLM), chest HRCT image

## Abstract

Chronic obstructive pulmonary disease (COPD) is a preventable, treatable, progressive chronic disease characterized by persistent airflow limitation. Patients with COPD deserve special consideration regarding treatment in this fragile population for preclinical health management. Therefore, this paper proposes a novel lung radiomics combination vector generated by a generalized linear model (GLM) and Lasso algorithm for COPD stage classification based on an auto-metric graph neural network (AMGNN) with a meta-learning strategy. Firstly, the parenchyma images were segmented from chest high-resolution computed tomography (HRCT) images by ResU-Net. Second, lung radiomics features are extracted from the parenchyma images by PyRadiomics. Third, a novel lung radiomics combination vector (3 + 106) is constructed by the GLM and Lasso algorithm for determining the radiomics risk factors (K = 3) and radiomics node features (d = 106). Last, the COPD stage is classified based on the AMGNN. The results show that compared with the convolutional neural networks and machine learning models, the AMGNN based on constructed novel lung radiomics combination vector performs best, achieving an accuracy of 0.943, precision of 0.946, recall of 0.943, F1-score of 0.943, and ACU of 0.984. Furthermore, it is found that our method is effective for COPD stage classification.

## 1. Introduction

As a common and non-infectious lung disease, chronic obstructive pulmonary disease (COPD) presents a preventable, treatable, and progressive chronic disease with debilitating lung conditions characterized by persistent airflow limitation [1]. Severe COPD can cause chronic morbidity and eventually lead to death. In 2030, it will become the third-largest death factor worldwide [2].

Because of this characterization, the COPD stage is diagnosed from stage 0 to IV according to Global Initiative for Chronic Obstructive Lung Disease (GOLD) criteria accepted by the American Thoracic Society and the European Respiratory Society [1,3]. The assessment parameters in the GOLD criteria are the forced expiratory volume in 1 s/forced vital capacity (FEV_1_/FVC) and FEV_1_ % predicted [1,4]. The FEV_1_/FVC and FEV_1_ % predicted can explain the impact on symptoms and life quality of COPD patients [5,6], but they cannot reflect the change of the lung tissue in COPD patients with COPD stage evolution. The predicted FEV_1_/FVC and FEV_1_ % changes only occur when lung tissue is destroyed to a certain extent. However, the measurement accuracy of PFT is limited by the compliance degree of patients. Specifically, the measurement process of PFT is very complex, and it is difficult for patients to understand and comply with the requirements put forward by doctors [7]. In addition, PFT cannot intuitively provide detailed anatomical information and morphological changes, such as subtypes of emphysema and bronchial wall thickening [8,9].

Compared with the GOLD criteria and other imaging equipment, computed tomography (CT) has been regarded as the most effective modality for characterizing and quantifying COPD [10]. For example, compared with PFT, chest CT images can indicate the patients have suffered from mild lobular central emphysema and decreased exercise tolerance in smokers without airflow limitation [11]. In addition, with the significant progress of CT imaging, especially high-resolution CT (HRCT), it has become an effective method for quantitative analysis of COPD, such as measuring the severity of air trapping, emphysema, airflow obstruction, and airway diseases [12,13]. However, the quantitative analysis of the bronchial and vascular flow are limited by the resolution of the HRCT. Therefore, it is challenging to automatically, semi-automatically, or manually segment the tiny trachea (small airway) and vascular from chest CT images. In particular, small airways (diameter <2 mm) and associated vessels can hardly be observed from chest CT images. Based on the above, parametric response mapping (PRM) [14] was proposed to locate the small airway lesion region, the emphysema region, and the healthy lung region. First, PRM needs to register the expiratory and inspiratory chest CT images. Then, the mall airway lesion, the emphysema, and the healthy lung region are located by the set thresholds. Therefore, the registration method directly affects the location of the above regions.

Radiomics features in lung disease imaging have been regarded as the state of the art for clinicians [15]. However, radiomics features in COPD develop slower than in other lung diseases, such as lung cancer and pulmonary nodules. The diffuse distribution of COPD in the lung limits the application of radiomics features in COPD. Radiomics features should be extracted from the region of interest (ROI) of the chest CT images. However, the diffuse distribution of COPD makes it difficult to determine ROI. Until 2020, Refaee T. et al. point out that radiomics features have not been extensively investigated yet in COPD [16]. There are potential applications of radiomics features in COPD for the diagnosis, treatment, and follow-up of COPD and future directions [16]. Meanwhile, the value of lung radiomics features in COPD assessment has also been confirmed [17].

Lung radiomics features extracted from the peripheral airway, pulmonary parenchyma, and pulmonary vessels in the chest CT images are suitable for reflecting the change of the lung tissue in COPD patients with COPD stage evolution. Specifically, the characteristic pathological changes of COPD exist in the central airway (trachea, bronchus, bronchioles with an inner diameter greater than 2–4 mm), peripheral airway (bronchioles and bronchioles with an inner diameter less than 2 mm), and pulmonary parenchyma and pulmonary vessels. Chronic inflammation causes the airway wall to repeatedly be damaged and repaired as the COPD stage evolves, resulting in airway blockage and a narrow air cavity in the peripheral airway [18]. In addition, the destruction of pulmonary parenchyma in patients with COPD involves the expansion and collapse of respiratory bronchioles. With COPD stage evolution, this expansion and collapse of respiratory bronchioles spread from the upper region to the whole lung, and the pulmonary capillary beds destroy [19]. In the earlier COPD stages, the changes in pulmonary vessels are characterized by the thickening of the vascular wall. With the continuous development of COPD, the increase of smooth muscle, proteoglycan, and collagen further thickens pulmonary vessels’ walls, which may lead to cor pulmonale [20]. Therefore, COPD results from the joint action of the peripheral airway, pulmonary parenchyma, and pulmonary vessels. Thus, the peripheral airway, pulmonary parenchyma, and pulmonary vessels as ROI [17] to extract lung radiomics features are reasonable for COPD stage classification.

Currently, radiomics features have also been used in COPD for survival prediction [21,22], spirometric assessment of emphysema presence and severity [23], COPD exacerbations [24], COPD early decision [3], COPD stage classification [25,26], COPD prediction [27,28], and analysis of COPD and resting heart rate [29]. The convolutional neural networks (CNN) and machine learning (ML) models can implement the COPD stage classification task. Compared with the CNN based on chest HRCT images, the multi-layer perceptron (MLP, a kind of ML model) classifier performs better, achieving an accuracy of 0.83, precision of 0.83, recall of 0.83, F1-score of 0.82, and AUC of 0.95 [25]. However, the classification performance needs to be further improved. The graph neural network (GNN) was first proposed in 2005 [30]. The GNN also developed into the graph convolutional network (GCN) inspired by CNN [31]. However, the fixed-graph structure of the GCN by using the entire dataset limits its application and development. To overcome the limitation of the GCN and maintain the advantages of the GNN, an auto-metric graph neural network (AMGNN) based on a meta-learning strategy is proposed [32]. The proposed AMGNN has been applied to the Alzheimer’s disease classification, achieving a good performance [32,33]. However, the risk factors and node features are critical for the classification performance of the AMGNN. Compared with node features, the risk factors with a lower dimension. The selection of the risk factors may have an important impact on the classification effect. Therefore, we focus on COPD and construct a novel lung radiomics combination vector based on the AMGNN for COPD stage classification.

Lung radiomics features have been applied to COPD stage classification, and compared to the CNN models, the ML models with lung radiomics features selected by least absolute shrinkage and selection operator (Lasso) algorithm perform better [25]. However, the risk factors and node features of the AMGNN limit the application in COPD stage classification. Therefore, this paper applies the AMGNN with lung radiomics features to improve COPD stage classification performance. Our contributions in this paper are briefly described as follows:

(1) This paper proposes a lung radiomics features selection method for risk factors and node features of the AMGNN with the advantages of modeling the correlation between samples and building a small graph structure to classify the COPD stage. The lung radiomics features are only extracted from routine chest HRCT images, eliminating the limitation that risk factors and node features based on prior knowledge are difficult to obtain, such as gene information. The Lasso algorithm with 10-fold cross-validation selects the independent variables related to the dependent variables, determining the critical, independent variables to simplify the classification model. The Lasso algorithm for improving COPD classification has been confirmed [25]. The Cox model [3] and the generalized linear model (GLM) are often used for determining the risk factors. Compared with the Cox model, the GLM eliminates the limitation of follow-up features with multiple time series. Last, a novel lung radiomics combination vector (3 + 106) is constructed by GLM and Lasso algorithm for determining the radiomics risk factors (K = 3) and radiomics node features (d = 106) of the AMGNN;

(2) Compared with previous work on COPD identification (binary classification: COPD and without COPD) [34,35], our work (COPD stage classification) has more clinical significance. Therefore, our proposed model eliminates the limitations of PFT and may become an effective tool for COPD management.

## 2. Materials and Methods

Materials and methods are described in detail in Section 2.1 and Section 2.2, respectively.

### 2.1. Materials

Figure 1 shows the participants’selection flow diagram and GOLD distribution of the participants in this study. Specifically, Figure 1a shows the participants’ selection flow. Our study cohort was enrolled in the national clinical research center of respiratory diseases, China, from 25 May 2009, to 11 January 2011. Four hundred and sixty-five Chinese participants aged 40–49 were included in this study after being strictly selected by the inclusion and exclusion criteria [36]. Then, these 465 participants underwent chest HRCT scans (TOSHIBA, kVp: 120 kV, X-ray tube current: 40 mA, slice thickness: 1.0 mm) at the deep inspiration state and PFT on the same day. The COPD stage 0-III-IV (GOLD 0-III-IV) is diagnosed by GOLD 2008 using FEV_1_/FVC and FEV_1_% predicted PFT [1,4]. Figure 1b shows that our study cohort has 129, 108, 121, and 107 participants in the GOLD 0, I, II, and III-IV, respectively. The 465 participants are divided into the train set (70%) and the test set (30%). Furthermore, Figure 1c,d shows the detailed training and test sets in each COPD stage.

This study was approved by the ethics committee in the national clinical research center of China’s respiratory diseases. In addition, all participants have provided written informed consent to the first affiliated hospital of Guangzhou Medical University before chest HRCT scans and PFT.

### 2.2. Methods

Figure 2 shows the detailed flow chart of our proposed method in this study.

#### 2.2.1. Lung Parenchyma Segmentation and Radiomics Feature Extraction

Figure 2A(a) shows that a state-of-the-art ResU-Net (U-net (R231)) [37] trained by data diversity (diverse lung disease images), which has been a robust and standard segmentation model of pathological lungs, is transferred to segment lung parenchyma images from 465 sets of chest HRCT images. The network architecture of the ResU-Net has been described in detail in our previous study [38]. Then, Figure 2A(b) shows that 1316 lung radiomics features of each participant are extracted from lung parenchyma images with the Hounsfield unit [39] by PyRadiomics [40].

Figure 2B shows the lung radiomics feature extraction model: PyRadiomics. Specifically, two steps are performed to extract the lung radiomics features from lung parenchyma images. The two steps include (1) original lung parenchyma images are filtered by wavelet and Laplacian of Gaussian (LoG) filters, generating derived lung parenchyma images; (2) the original and derived lung parenchyma images are further used to calculate the lung radiomics features based on the preset classes. More detailed descriptions can be found in our previous studies [25,29].

#### 2.2.2. Radiomics Feature Combination

The risk factors and node features are critical for the classification performance of the AMGNN. Therefore, a novel lung radiomics combination vector (3 + 106) is constructed by the GLM [41] and the Lasso algorithm [42] for determining the radiomics risk factors (K = 3) and radiomics node features (d = 106) of the AMGNN.

The results of a previous study [25] have shown that the Lasso algorithm helps improve the classification effect of COPD. Specifically, Figure 2A(c) shows that the Lasso algorithm with 10-fold cross-validation selects the radiomics node features from 1316 lung radiomics features. Meanwhile, the radiomics risk factors are selected from 1316 lung radiomics features by calculating the top three maximal R^2^ values generated from the GLM. Finally, the radiomics risk factors (K = 3) and radiomics node features (d = 106) are concatenated, obtaining the proposed lung radiomics combination vector (3 + 106) for COPD classification.

Equations (1) and (2) give the mathematical form of the GLM and Lasso [3,25,29], respectively.
(1)$$g(\mu_{y_{i}})=\beta_{0}+\sum_{j=1}^{p}\beta_{j}\cdot x_{j},$$
where the link function $g(\mu_{y_{i}})=\eta$ relates the mean $\mu_{y_{i}}=E(y_{i})$ to the linear predictor $\eta =\sum_{j=1}^{p}\beta_{j}\cdot x_{j}$. *y_i_* denotes the dependent variable (the 465 COPD stages: GOLD 0, I, II, and III-IV). *x**_j_* denotes the independent variable (the 465 × 1316 normalized lung radiomics features). *β_j_* denotes the regression coefficients *i*∈[1, n] and *j*∈[0, *p*].
(2)$$A\leftarrow arg\;min\left\{\sum_{i=1}^{n}{\left(y_{i}-\beta_{0}-\sum_{j=1}^{p}\beta_{j}x_{j}\right)}^{2}+\lambda \sum_{j=0}^{p}\left|\beta_{j}\right|\right\}$$
where matrix *A* denotes the node features (selected lung radiomics feature). *x**_j_* denotes the independent variable (the 465 × 1316 normalized lung radiomics features). *y_i_* denotes the independent variable (the 465 COPD stages: GOLD 0, I, II, and III–IV). *λ* denotes the penalty parameter (*λ* ≥ 0). *β_j_* denotes the regression coefficients *i*∈[1, n] and *j*∈[0, *p*].

#### 2.2.3. COPD Stage Classification Based on AMGNN

Figure 2A(d) shows that the COPD stage is classified based on AMGNN with the proposed lung radiomics combination vector. Specifically, Figure 2C shows the pipeline of AMGNN based on meta-learning. Specifically, T graphs are separately established by selecting the 40 known nodes in the training set and 1 unknown node in the training set. The 40 (10 × 4) known nodes of each graph include 10 known nodes in each COPD stage, and four COPD stages are included in our study. Subsequently, T-1 meta tasks train the AMGNN by randomly to the constantly updated network parameter P and loss value ℓ, establishing the AMGNN (P_T_). Lastly, the established AMGNN (P_T_) realizes the unknown nodes in the test set for COPD stage classification.

In addition, Figure 2D shows the detailed network structure of AMGNN (*P_i_*) for calculating the auto-metics connection with probability constraints and updating the node. Specifically, two stages are parallelly performed in each AMGNN (*P_i_*) for generating the adjacency matrix $\tilde{A}$. First, the adjacency matrix $\tilde{A}$ multiplies the proposed lung radiomics combination vector of the nodes in the *l^th^* layer $V^{\left(l\right)}$, and then updating the node $Gn(V^{\left(l\right)})$ is obtained by a nonlinear activation function FC (Leaky ReLU). Finally, the output $V^{\left(l+1\right)}$ of the AMGNN (*P_i_*) is obtained by concatenating $V^{\left(l\right)}$ and $Gn(V^{\left(l\right)})$. However, the output of the last AMGNN (*P_T-1_*) is input into the softmax instead of the concatenating operation.

Furthermore, one stage is constructing the edge weight matrix W using the 106 radiomics nodes features. Meanwhile, the other stage is constructing the edge constraint matrix E using the 3 radiomics risk factors. Finally, the edge weight matrix W element wisely multiplies the edge constraint matrix E, obtaining the adjacency matrix $\tilde{A}$. In constructing the W stage, the N × N × 106 feature block C is generated by copying N times of the N × 106 feature map. Then, each N × N × 1 in the N × N × 106 feature block C is transposed, generating a new N × N × 106 feature block C′. The difference feature block C′′ is further obtained by calculating the absolute difference between each feature of the two nodes in C and C′. Subsequently, a CNN with a 1 × 1 kernel generates the edge weight matrix W. The detailed description of calculating the edge constraint matrix E refers to the study [32].

## 3. Experiments and Results

### 3.1. Experiments

Figure 3 shows four experiments to verify the effectiveness of our proposed method. The radiomics/CNN combination vectors (K risk factors + d node features) based on the AMGNN for COPD classification in Experiments 2–4 can be obtained in Appendix A. The open-source code of AMGNN with 5-fold cross-validation is directly applied to COPD stage classification. Similarly, other ML models are also trained with the 5-fold cross-validation. Finally, the trained models with the best AUC are loaded and used on the test set.

The CNN models based on the chest HRCT images perform unsatisfactory classification [25]. However, the model fusion strategy based on the CNN and ML models should be further studied. Specifically, the features are extracted by the CNN model based on transfer learning, and then the ML classifiers are used for classification. Therefore, 3D CNN features are extracted based on the encoder backbone (ResNet 10) of a pre-trained Med3d [41] using truncated transfer learning for comparing the lung radiomics features. Med3d, a heterogeneous 3D network, is used to extract general medical 3D features by building a 3DSeg-8 dataset with diverse modalities, target organs, and pathologies. Five hundred and twelve 3D CNN feature maps with the size of 3 × 3 × 3 are generated by ResNet 10. Therefore, each participant has 13,824 3D CNN features (512 × 3 × 3 × 3 = 13,824) by flattening the 512 3D CNN feature maps.

Experiments 1 is designed based on ML classifiers with lung radiomics or 3D CNN features. The support vector machine (SVM) [42], multi-layer perceptron (MLP) [25,43], random forest (RF) [44], logistic regression (LR) [45], and the proposed AMGNN are applied to diagnose Alzheimer’s disease [32]. In addition, SVM, LR, MLP, RF, gradient boosting (GB) [46], and linear discriminant analysis (LDA) [47] are applied to classify the COPD stages [25]. Therefore, the above six classic ML classifiers and the AMGNN are used for COPD stage classification. Specifically, the original lung radiomics (1316) and 3D CNN features (13,824) are directly and separately used to classify the COPD stage based on ML classifiers. In addition, the radiomics features (106) and 3D CNN features (60) of each participant are automatically selected by the Lasso algorithm with the 10-fold cross-validation. For comparison of the same number of features, the radiomics features (106) and 3D CNN features (60) of each participant are separately selected or fused by the GLM/principal component analysis (PCA) algorithm. These selected or fused features are separately used to classify the COPD stage based on ML classifiers. However, the combination feature vector of the lung radiomics or 3D CNN features selected by GLM/Lasso and these features separately fused by PCA are also considered in our experimental design in Experiment 1. Specifically, because the GLM and Lasso algorithm perform the feature selection task, the features selected by the GLM and Lasso algorithm are not further combined.

Experiments 2–4 are designed based on the AMGNN with lung radiomics or 3D CNN features for determining the risk factors and node features. Specifically, compared with node features of AMGNN, the risk factors have a lower dimension. Therefore, the PCA algorithm and GLM are separately used to determine the risk factors (K = 2–6). In addition, the PCA algorithm and GLM are separately used to determine the risk factors (K = 2–6) based on the original lung radiomics (1316) or 3D CNN features (13,824), and the corresponding node features are original lung radiomics (1316) or 3D CNN features (13,824) in Experiment 2. The PCA and GLM are separately used to determine the risk factors (K = 2–6) based on the selected lung radiomics (106) or 3D CNN features (60) by the Lasso algorithm, and the corresponding node features also are the selected lung radiomics (106) or 3D CNN features (60) in Experiment 3. The PCA and GLM are separately used to determine the risk factors (K = 2–6) based on 1316 original lung radiomics or 13,824 3D CNN features, and the corresponding node features are the lung radiomics (106) or 3D CNN features (60) selected by the Lasso algorithm in Experiment 4.

Table 1 reports the different ML classifiers with their definitions. The 4-way-10-shot of the few-shot learning [48] is set in the AMGNN (iterations = 600, batch_size_test = 28, batch_size_train = 28, and random_seed = 42). All the source codes run on PyCharm 2020.3.5 (professional edition) on Windows 10 Pro 64-bit with two 2080 Ti GPUs, 32 GB RAM, 1 TB mechanical storage, and a 256 G SSD.

### 3.2. Results

This section reports the results of Experiments 1-4, including the five standard evaluation metrics (accuracy, precision, recall, F1-score, and area under the curve (AUC)). The evaluation metric AUC for multi-classification is calculated by the receiver operating characteristic curve (ROC) [25]. Compared with the CNN and ML models, the AMGNN based on the novel lung radiomics combination vector (3 + 106) performs best, achieving an accuracy of 0.943, precision of 0.946, recall of 0.943, F1-score of 0.943, and ACU of 0.984.

#### 3.2.1. COPD Stage Classification Based on Different ML Classifiers

Table 2, Table 3, Table 4, Table 5, Table 6 and Table 7 report the ML classifiers’ performance in Experiment 1. Meanwhile, Figure 4 and Figure 5 visually show the evaluation metrics and ROC curves of the ML classifier in Experiment 1. The MLP classifier performs better than the other ML classifiers for COPD stage classification. Specifically, the MLP classifier with 13,824 original 3D CNN/1316 original lung radiomics features performs best, achieving an accuracy of 0.793/0.786, precision of 0.798/0.784, recall of 0.793/0.784, F1-score of 0.790/0.784, and ACU of 0.938/0.919. In addition, 60 3D CNN/106 lung radiomics features separately selected or fused by the Lasso algorithm, GLM, and PCA algorithm are further used to classify the COPD stage. Again, the MLP classifier with 60 3D CNN/106 lung radiomics features selected by the Lasso algorithm performs best, achieving an accuracy of 0.821/0.829, precision of 0.826/0.828, recall of 0.821/0.829, F1-score of 0.821/0.824, and ACU of 0.946/0.950. Even for the CNN/radiomics combination vector, the MLP classifier with 60 3D CNN/106 lung radiomics features only selected by the Lasso algorithm also performs best.

The GLM fails to improve the classification performance of all ML classifiers with 3D CNN features. Compared with the GLM, the PCA algorithm only improves the classification performance of the RF classifier with 3D CNN features. The Lasso algorithm only improves the classification performance of the MLP classifier with the 3D CNN features. In addition, the GLM fails to improve the classification performance of the SVM classifier, but it improves the classification performance of other ML classifiers with lung radiomics features. Compared with the GLM, the PCA algorithm only improves the classification performance of the RF and LDA classifiers with lung radiomics features. However, the Lasso algorithm improves the classification performance of all ML classifiers with lung radiomics features.

Compared to the evaluation metrics of the MLP classifier with 13,824 original 3D CNN features, that of the MLP classifier with 60 selected 3D CNN features has been improved by 2.8% in accuracy, 2.8% in precision, 2.8% in recall, 3.1% in F1-score, and 0.8% in ACU. Similar to 3D CNN features, compared to the evaluation metrics of the MLP classifier with 1316 original lung radiomics features, that of the MLP classifier with 106 selected lung radiomics features has been improved by 4.3% in accuracy, 4.4% in precision, 4.5% in recall, 4.0% in F1-score, and 3.1% in ACU.

#### 3.2.2. COPD Stage Classification Based on the AMGNN Classifier

Table 8, Table 9 and Table 10 report the AMGNN classifier’s performance in Experiments 2–4. Meanwhile, Figure 6 and Figure 7 visually show the evaluation metrics and ROC curves of the AMGNN classifier in Experiments 2–4. The AMGNN classifier with the proposed lung radiomics combination vector constructed by 3 radiomics risk factors (selected from 1316 original lung radiomics features by GLM) and 106 radiomics node features (selected from 1316 original lung radiomics features by the Lasso algorithm) performs best for COPD stage classification, achieving an accuracy of 0.943, precision of 0.946, recall of 0.943, F1-score of 0.943, and ACU of 0.984.

The results in this section show that the AMGNN classifier performs better than the ML classifiers. Specifically, compared to the best evaluation metrics of the MLP classifier with 13,824 original 3D CNN features in Experiment 1, that of the AMGNN classifier with 3D CNN combination vector (K = 6 (GLM) + d = 13,824) in Experiment 2 has improved by 4.3% in accuracy, 4.2% in precision, 4.3% in recall, 4.1% in F1-score, and 1.2% in ACU. Meanwhile, compared to the best evaluation metrics of the MLP classifier with 1316 original lung radiomics features in Experiment 1, that of the AMGNN classifier with lung radiomics combination vector (K = 5 (PCA) + d = 1316) in Experiment 2 has improved by 11.4% in accuracy, 11.5% in precision, 11.6% in recall, 11.6% in F1-score, and 5.3% in ACU. Compared to the best evaluation metrics of the MLP classifier with 60 3D CNN features selected by the Lasso algorithm in Experiment 1, that of the AMGNN classifier with 3D CNN combination vector (K = 4 (PCA) + d = 60 in Experiment 3/K = 3 (PCA) + d = 60 in Experiment 4) has improved by 1.5%/2.2% in accuracy, 0.2%/2.5% in precision, 1.5%/2.2% in recall, 1.2%/2.0% in F1-score, and 1.1%/1.2% in ACU, respectively. Meanwhile, compared to the best evaluation metrics of the MLP classifier with 106 lung radiomics features selected by the Lasso algorithm in Experiment 1, that of the AMGNN classifier with lung radiomics combination vector (K = 2 (PCA) + d = 106 in Experiment 3/K = 3 (GLM) + d = 106 in Experiment 4) in Experiments 4–5 has improved by 10.0%/11.4% in accuracy, 10.1%/11.8% in precision, 10.0%/11.4% in recall, 10.4%/11.9% in F1-score, and 3.4%/3.4% in ACU, respectively.

Table 8 and Table 10 and Figure 6 and Figure 7 show that the Lasso algorithm helps the AMGNN classifier to construct the excellent edge weight matrix by reducing redundant collinearity 3D CNN or lung radiomics features. The mean evaluation metrics of the CNN/radiomics combination vector (K = 2–6 (PCA/GLM) + d = 60/106) in Experiment 4 also perform better than that of the CNN/radiomics combination vector (K = 2–6 (PCA/GLM) + d = 13,824/1316) in Experiment 2 based on the AMGNN classifier. Specifically, compared with the mean evaluation metrics of the CNN/radiomics combination vector (K = 2–6 (PCA/GLM) + d = 13,824/1316) in Experiment 3, that of the CNN/radiomics combination vector (K = 2–6 (PCA/GLM) + d = 60/106) in Experiment 4 has improved by 1.4%/0.2%/2.3%/3.0% in accuracy, 0.5%/0.4%/3.3%/3.1% in precision, 1.4%/0.2%/2.3%/3.0% in recall, 2.2%/1.1%/2.5%/3.2% in F1-score, and 0.9%/1.8%/1.7%/1.3% in ACU, respectively.

Table 9 and Table 10 and Figure 6 and Figure 7 show that the PCA/GLM helps the AMGNN classifier determine the risk factors for constructing the excellent edge constraint matrix. The mean evaluation metrics of the CNN/radiomics combination vector (K = 2–6 (PCA/GLM) + d = 60/106, K risk factors separately generated by PCA/GLM from 13,824 original 3D CNN/1316 original lung radiomics features) in Experiment 4 also perform better than that of the CNN/radiomics combination vector (K = 2–6 (PCA/GLM) + d = 13,824/1316, K risk factors separately generated by PCA/GLM from 60 selected 3D CNN/106 selected lung radiomics features) in Experiment 3 based on the AMGNN classifier. Specifically, compared with the mean evaluation metrics of the CNN/radiomics combination vector (K = 2–6 (PCA/GLM) + d = 13,824/1316) in Experiment 3, that of the CNN/radiomics combination vector (K = 2–6 (PCA/GLM) + d = 60/106) in Experiment 4 has improved by –0.4%/0.5%/0.9%/1.4% in accuracy, –0.7%0.6/1.4%/1.4% in precision, –0.4%/0.5%/0.9%/1.4% in recall, –0.4%/0.6%/0.9%/1.4% in F1-score, and –1.9%/0.5%/1.0%/1.1% in ACU, respectively.

Table 8, Table 9 and Table 10 and Figure 6 and Figure 7 also show that the evaluation metrics of the AMGNN classifier with lung radiomics combination vector are better than the AMGNN classifier with the 3D CNN combination vector.

Specifically, the AMGNN classifier with the 3D CNN combination vector (K = 4 or 5/6+ d = 13,824, K risk factors separately generated by PCA/GLM from 13,824 original 3D CNN features) performs best in Experiment 2, achieving an accuracy of 0.807 or 0.807/0.836, precision of 0.831 or 0.811/0.840, recall of 0.807 or 0.807/0.836, F1-score of 0.799 or 0.804/0.831, and ACU of 0.929 or 0.924/0.840. Meanwhile, the AMGNN classifier with the lung radiomics combination vector (K = 5/5+ d = 13,824, K risk factors separately generated by PCA/GLM from 1316 original lung radiomics features) performs best in Experiment 2, achieving an accuracy of 0.900/0.871, precision of 0.899/0.871, recall of 0.900/0.871, F1-score of 0.900/0.870, and ACU of 0.972/0.964. Compared to the best evaluation metrics of the 3D CNN combination vector (K = 6 (GLM) + d = 13,824), the best evaluation metrics of the lung radiomics combination vector (K = 5 (PCA) + d = 13,824) have improved by 6.4% in accuracy, 5.9% in precision, 6.4% in recall, 6.9% in F1-score, and 2.2% in ACU.

Specifically, the AMGNN classifier with the 3D CNN combination vector (K = 4/4+ d = 60, K risk factors separately generated by PCA/GLM from 60 selected 3D CNN features) performs best in Experiment 3, achieving an accuracy of 0.836/0.814, precision of 0.846/0.837, recall of 0.836/0.814, F1-score of 0.833/0.812, and ACU of 0.957/0.949. Meanwhile, the AMGNN classifier with the lung radiomics combination vector (K = 3/3 or 6+ d = 106, K risk factors separately generated by PCA/GLM from 106 selected lung radiomics features) performs best in Experiment 3, achieving an accuracy of 0.907/0.886 or 0.886, precision of 0.914/0.887 or 0.886, recall of 0.907/0.886 or 0.886, F1-score of 0.908/0.886 or 0.881, and ACU of 0.983/0.956 or 0.963. Compared to the best evaluation metrics of the 3D CNN combination vector (K = 4 (PCA) + d = 60), the best evaluation metrics of the lung radiomics combination vector (K = 3 (PCA) + d = 106) have improved by 7.1% in accuracy, 6.8% in precision, 7.1% in recall, 7.5% in F1-score, and 2.6% in ACU.

Specifically, the AMGNN classifier with the 3D CNN combination vector (K = 3/6+ d = 60, K risk factors separately generated by PCA/GLM from 13,824 original 3D CNN features) performs best in Experiment 4, achieving an accuracy of 0.843/0.821, precision of 0.851/0.833, recall of 0.843/0.821, F1-score of 0.841/0.818, and ACU of 0.958/0.945. Meanwhile, the AMGNN classifier with the lung radiomics combination vector (K = 3/3 or 6+ d = 106, K risk factors separately generated by PCA/GLM from 1316 original lung radiomics features) performs best in Experiment 4, achieving an accuracy of 0.929/0.943, precision of 0.929/0.946, recall of 0.929/0.943, F1-score of 0.928/0.943, and ACU of 0.984/0.984. Compared to the best evaluation metrics of the 3D CNN combination vector (K = 3 (PCA) + d = 60), the best evaluation metrics of the lung radiomics combination vector (K = 3 (GLM) + d = 106) have improved by 10.0% in accuracy, 9.5% in precision, 10.0% in recall, 10.2% in F1-score, and 2.6% in ACU.

Table 11 compares the results of our proposed method with other previous methods. Furthermore, it fully demonstrates the superiority of our method over other previous methods.

## 4. Discussion

Based on the experimental results, we give the following discussion and point out the limitations in this study and the future direction. Compared with multimodal sleep data [49] and electromyography [50], lung radiomics features extracted from chest HRCT images are more suitable for COPD stage classification.

First, the MLP classifier with 3D CNN or lung radiomics features performs best for the COPD stage classification in the ML classifiers. The MLP classifier’s structure and ability to handle complex nonlinear features determine its excellent ability. The MLP classifier is composed of three full connection layers, which is more efficient and more suitable for modeling long-range dependencies [51]. Meanwhile, COPD patients have high heterogeneity and different phenotypes [1], resulting in complex nonlinear 3D CNN or lung radiomics features extracted from their chest HRCT images. However, the MLP classifier can handle these complex nonlinear features and discover dependencies between different input features by approximating the nonlinear map globally to realize the COPD stage classification [25,43].

Second, the AMGNN classifier performs better than the best MLP classifier in the COPD stage classification. This is because the AMGNN classifier, which overcomes the lack of flexibility of existing GNN models, introduces a meta-learning strategy, and is insensitive to graphics size [32]. In addition, the AMGNN classifier needs fewer training samples than traditional ML classifiers while maintaining stability in reducing training samples [32]. Therefore, even if a small number of graphs are used, it also can show good performance for the COPD stage classification. The AMGNN classifier also solves the problems that the standard medical image cohort is usually challenging to obtain, which results in a small data volume. On the other hand, the MPL classifier needs sufficient data to train its parameters. Therefore, the MPL classifier’s performance with small data may deteriorate very seriously. Meanwhile, the AMGNN classifier performs well on the test set, showing no underfitting and overfitting problems. Specifically, too deep/complex network structure often causes underfitting. However, the AMGNN classifier only with two layers effectively avoids underfitting. In addition, the AMGNN classifier introduces the meta-learning strategy. Meta-learning-based approaches can increasingly train powerful CNNs on small datasets in many vision problems [52]. Therefore, the meta-learning strategy effectively avoids overfitting.

Third, the Lasso algorithm improves the MLP and AMGNN classifiers’ performance. Compared with the Lasso algorithm, the PCA algorithm reduces the original features’ dimension by identifying the orthogonal linear combinations with the largest covariance. Therefore, the PCA algorithm can compress the original features into a small number of features and obtain new fused features without losing too much information. However, because the PCA algorithm needs to retain most of the features, the fused features still affect the COPD classification effect. The Lasso algorithm and GLM perform the feature selection task. Compared with the GLM, the Lasso algorithm is a penalized likelihood approach. Therefore, the Lasso algorithm automatically selects the original features, obtaining the fixed numbers of the selected features. However, GLM generates R^2^ values corresponding to each feature instead of the fixed numbers of the selected features. In addition, the Lasso algorithm is often used with survival analysis models to determine variables and eliminate the collinearity problem between variables [3]. The Lasso algorithm is also applied to select the features to improve the MLP classifier’s performance by establishing the relationship between the independent variables (3D CNN or lung radiomics features) and dependent variables (the COPD stages) [25]. Therefore, the complexity of the 3D CNN or lung radiomics features is reduced. As a result, the MLP classifier can focus on the 60 selected 3D CNN or 106 selected lung radiomics features to improve the classifier’s performance. The Lasso algorithm also applies to the AMGNN classifier to construct the excellent edge weight matrix by reducing redundant collinearity 3D CNN or lung radiomics features. Meanwhile, the Lasso algorithm determines node features of the AMGNN classifier, eliminates collinearity features, and further avoids overfitting.

Fourth, lung radiomics features remove the limitation of risk factors and node features that are difficult to obtain. Meanwhile, the AMGNN classifier with the lung radiomics combination vector performs better than the 3D CNN combination vector. The risk factors of diseases, such as the risk factors (age, gender, year of education, and APOe4 gene information) of Alzheimer’s disease [32], are hardly obtained or determined, which limits the application of the AMGNN classifier. However, because a large number of lung radiomics features can be extracted from the ROI of chest HRCT images, we may overcome the limitation of obtaining and determining risk factors. Compared with the 3D CNN features, the lung radiomics features are calculated by preset formulas, which are easier to explain in subsequent studies.

Lastly, this study also has some limitations, and we point out the future direction. Although the AMGNN classifier with the proposed novel lung radiomics combination vector achieves promising results, the model must be retrained with a newcomer to be classified. Furthermore, we only explained the method from the engineering and algorithm, but the clinical significance of three radiomics risk factors needs further analysis by COPD experts and doctors. However, the combination vector of 3D CNN and lung radiomics based on ML or AMGNN classifiers for the COPD stage classification should be studied in the future.

## 5. Conclusions

This paper constructs a novel lung radiomics combination vector generated by GLM and Lasso algorithm for the COPD stage classification based on the auto-metric graph neural network. Compared with the CNN and ML models, the AMGNN based on the novel lung radiomics combination vector (K = 3 (GLM) + d = 106, 3 radiomics risk factors are selected by GLM and 106 radiomics node features are selected by the Lasso algorithm) performs best, achieving an accuracy of 0.943, precision of 0.946, recall of 0.943, F1-score of 0.943, and ACU of 0.984. Therefore, our proposed model eliminates the limitations of PFT and may become an effective tool for COPD management.

## 6. Patents

The method and device, electronic device and storage medium for stage classification of chronic obstructive pulmonary disease, CN2022104685981, Shenzhen Technology University and Northeastern University, China.

## Figures and Tables

**Figure 1 diagnostics-12-02274-f001:**
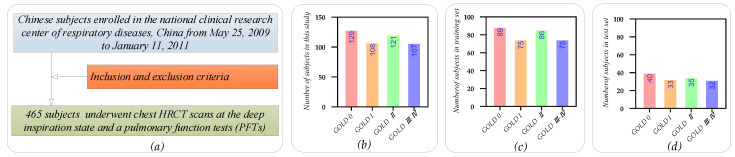
The participants’ selection flow diagram and GOLD distribution of the participants in this study. (**a**) The participants’ selection flow diagram; (**b**) GOLD distribution in our study cohort; (**c**) training set distribution; (**d**) test set distribution.

**Figure 2 diagnostics-12-02274-f002:**
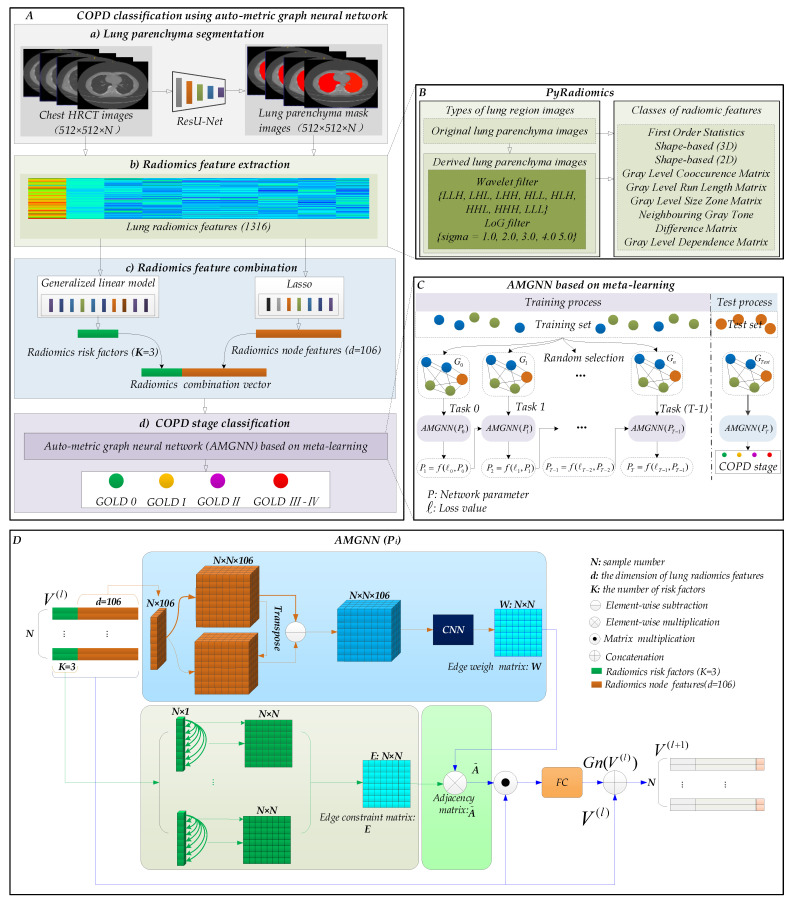
The proposed method in this study. (**A**) Our proposed method: COPD stage classification using auto-metric graph neural network; specifically, our proposed method includes (**a**) lung parenchyma segmentation, (**b**) radiomics feature extraction, (**c**) radiomics feature combination, and (**d**) COPD stage classification. (**B**) Lung radiomics feature extraction model: PyRadiomics [25,29,33,40]. (**C**) The pipeline of the AMGNN based on meta-learning. (**D**) Detailed network structure of the AMGNN (*P_i_*).

**Figure 3 diagnostics-12-02274-f003:**
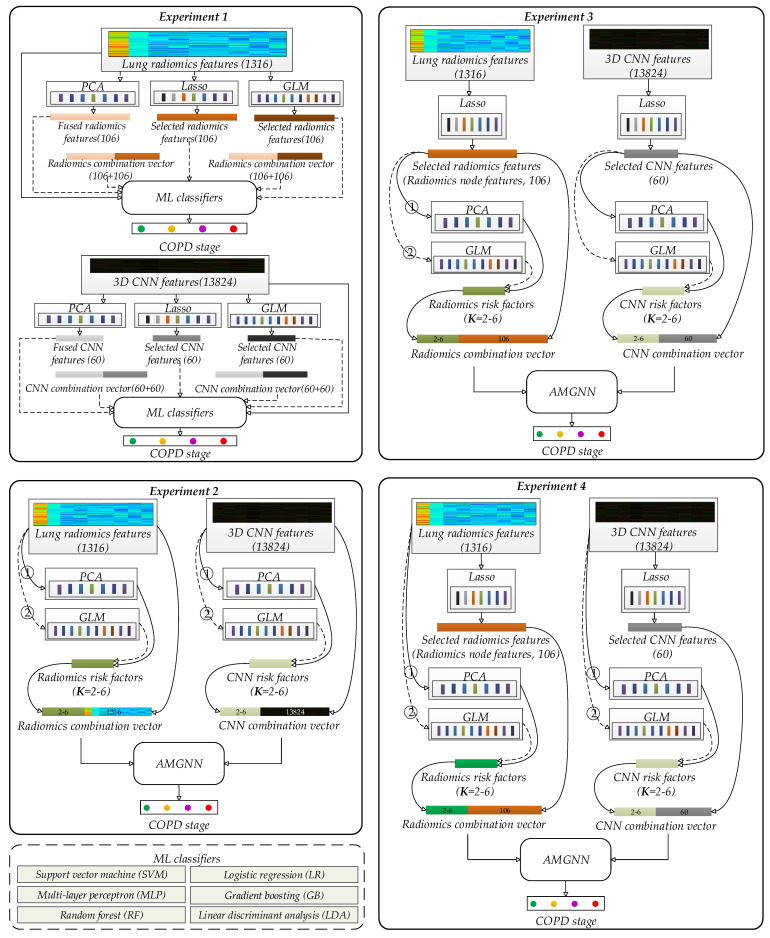
Experimental design in this paper.

**Figure 4 diagnostics-12-02274-f004:**
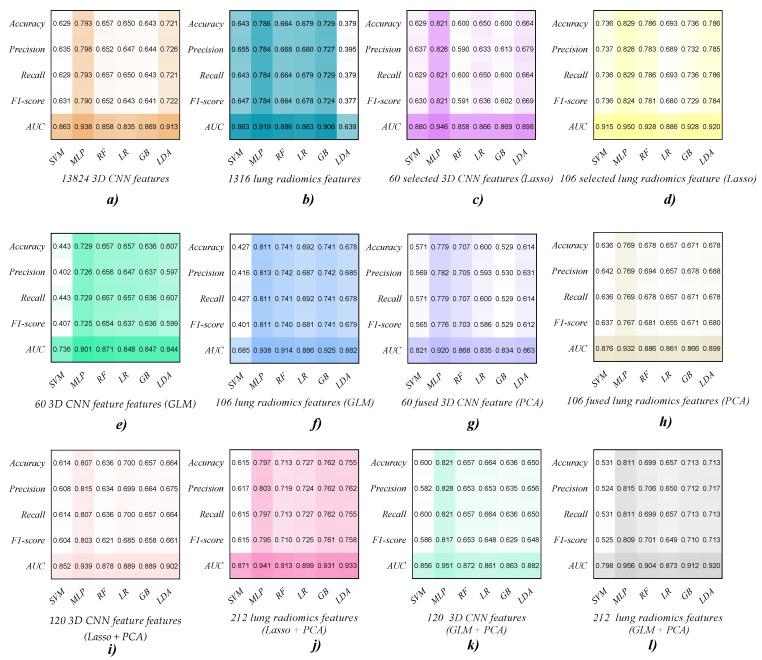
Visual evaluation metrics of different ML classifiers with different features in Experiment 1. (**a**,**b**) The evaluation metrics of different ML classifiers with 13,824 original 3D CNN/1316 original lung radiomics features. (**c**–**h**) The evaluation metrics of different ML classifiers with 60 3D CNN/106 lung radiomics features separately selected or fused by the Lasso algorithm, GLM, and PCA algorithm. (**i**–**l**) The evaluation metrics of different ML classifiers with different combinations of feature vectors of the selected and fused lung radiomics features or the selected and fused 3D CNN features.

**Figure 5 diagnostics-12-02274-f005:**
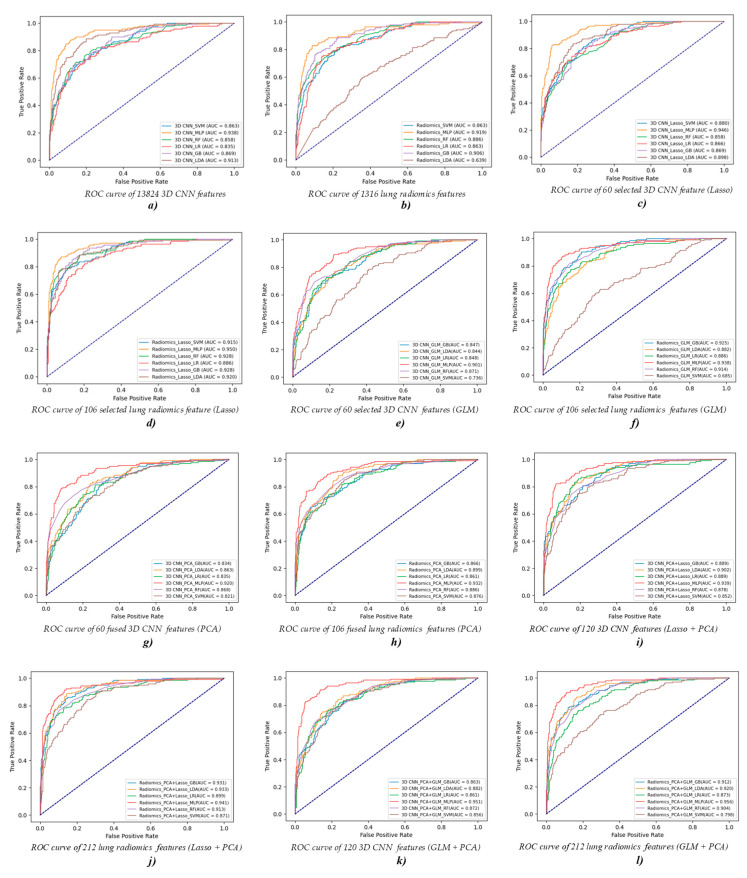
Visual ROC curves of different ML classifiers with different features in Experiment 1. (**a,b**) The ROC curves of different ML classifiers with 13,824 original 3D CNN/1316 original lung radiomics features. (**c**–**h**) The ROC curves of different ML classifiers with 60 3D CNN/106 lung radiomics features separately selected or fused by the Lasso algorithm, GLM, and PCA algorithm. (**i**–**l**) The ROC curves of different ML classifiers with different combinations of feature vectors of the selected and fused lung radiomics features or the selected and fused 3D CNN features.

**Figure 6 diagnostics-12-02274-f006:**
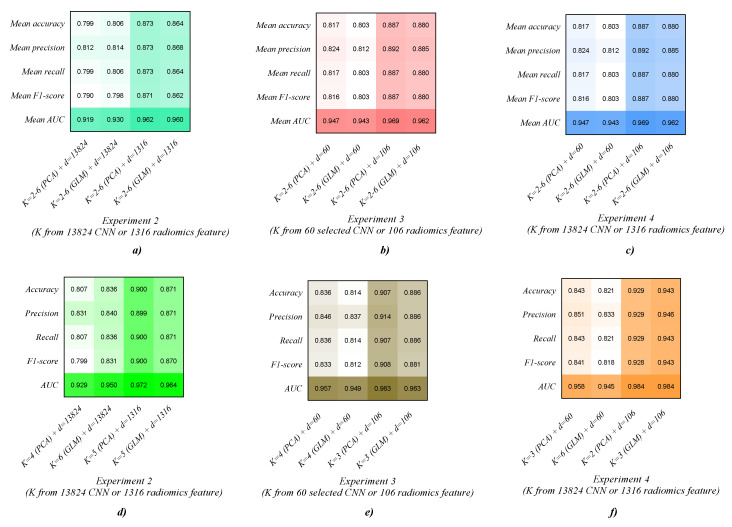
Visual evaluation metrics and ROC curves of the AMGNN classifier in Experiments 2–4. (**a**–**c**) The mean evaluation metrics, and (**d**–**f**) the best evaluation metrics.

**Figure 7 diagnostics-12-02274-f007:**
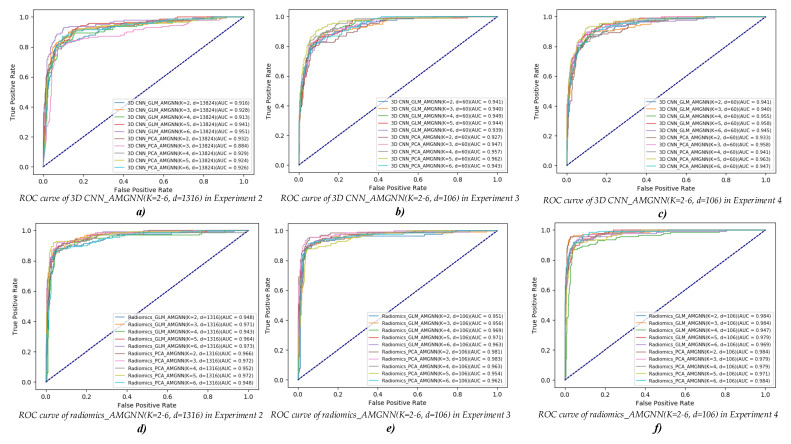
ROC curves of the AMGNN classifier in Experiments 2–4. (**a**–**f**) ROC curves of the AMGNN classifiers with different combination vectors.

**Table 1 diagnostics-12-02274-t001:** The different ML classifiers with their definitions.

ML Classifier	ML Classifier Model Definition in Python 3.6
SVM	SVM sklearn.svm.SVC(kernel=‘rbf’,probability=True)
MLP	sklearn.neural_network. MLPClassifier (hidden_layer_sizes=(400, 100), alpha=0.01, max_iter=10,000)
RF	sklearn.ensemble.RandomForestClassifier(n_estimators=200)
LR	sklearn.linear_model.logisticRegressionCV(max_iter=100,000, solver=“liblinear”)
GB	sklearn.ensemble.GradientBoostingClassifier()
LDA	sklearn.discriminant_analysis.()

**Table 2 diagnostics-12-02274-t002:** The different ML classifiers’ performance on the test set of original 3D CNN features (13,824)/lung radiomics features (1316) in Experiment 1.

Features	Classifier	Accuracy	Precision	Recall	F1-Score	AUC
3D CNN feature (13,824)/ Lung radiomics feature (1316)	SVM	0.629/0.643	0.635/0.655	0.629/0.643	0.631/0.647	0.863/0.863
	MLP	0.793/0.786	0.798/0.784	0.793/0.784	0.790/0.784	0.938/0.919
	RF	0.657/0.664	0.652/0.668	0.657/0.664	0.652/0.664	0.858/0.886
	LR	0.650/0.679	0.647/0.680	0.650/0.679	0.643/0.678	0.835/0.863
	GB	0.643/0.729	0.644/0.727	0.643/0.729	0.641/0.724	0.869/0.906
	LDA	0.721/0.379	0.726/0.395	0.721/0.379	0.722/0.377	0.913/0.639

**Table 3 diagnostics-12-02274-t003:** The different ML classifiers’ performance on the test set of selected 3D CNN features (60) or lung radiomics features (106) generated by the Lasso algorithm in Experiment 1.

Features	Classifier	Accuracy	Precision	Recall	F1-Score	AUC
Selected 3D CNN feature (60)/ Selected lung radiomics feature (106)	SVM	0.629/0.736	0.637/0.737	0.629/0.736	0.630/0.736	0.880/0.915
	MLP	0.821/0.829	0.826/0.828	0.821/0.829	0.821/0.824	0.946/0.950
	RF	0.600/0.786	0.590/0.783	0.600/0.786	0.591/0.781	0.858/0.928
	LR	0.650/0.693	0.633/0.689	0.650/0.693	0.636/0.680	0.866/0.886
	GB	0.600/0.736	0.613/0.732	0.600/0.736	0.602/0.729	0.869/0.928
	LDA	0.664/0.786	0.679/0.785	0.664/0.786	0.669/0.784	0.898/0.920

**Table 4 diagnostics-12-02274-t004:** The different ML classifiers’ performance on the test set of selected 3D CNN features (60) or lung radiomics features (106) generated by the GLM in Experiment 1.

Features	Classifier	Accuracy	Precision	Recall	F1-Score	AUC
Selected 3D CNN feature (60)/ Selected lung radiomics feature (106)	SVM	0.443/0.427	0.402/0.416	0.443/0.427	0.407/0.401	0.736/0.685
	MLP	0.729/0.811	0.726/0.813	0.729/0.811	0.725/0.811	0.901/0.938
	RF	0.657/0.741	0.656/0.742	0.657/0.741	0.654/0.740	0.871/0.914
	LR	0.657/0.692	0.647/0.687	0.657/0.692	0.637/0.681	0.848/0.886
	GB	0.636/0.741	0.637/0.742	0.636/0.741	0.636/0.741	0.847/0.925
	LDA	0.607/0.678	0.597/0.685	0.607/0.678	0.599/0.679	0.844/0.882

**Table 5 diagnostics-12-02274-t005:** The different ML classifiers’ performance on the test set of fused 3D CNN features (60) or lung radiomics features (106) generated by the PCA algorithm in Experiment 1.

Features	Classifier	Accuracy	Precision	Recall	F1-Score	AUC
Selected 3D CNN feature (60)/ Selected lung radiomics feature (106)	SVM	0.571/0.636	0.569/0.642	0.571/0.636	0.565/0.637	0.821/0.876
	MLP	0.779/0.769	0.782/0.769	0.779/0.769	0.776/0.767	0.920/0.932
	RF	0.707/0.678	0.705/0.694	0.707/0.678	0.703/0.681	0.868/0.886
	LR	0.600/0.657	0.593/0.657	0.600/0.657	0.586/0.655	0.835/0.861
	GB	0.529/0.671	0.530/0.678	0.529/0.671	0.529/0.671	0.834/0.866
	LDA	0.614/0.678	0.631/0.688	0.614/0.678	0.612/0.680	0.863/0.899

**Table 6 diagnostics-12-02274-t006:** The different ML classifiers’ performance on the test set of the CNN combination vector (Lasso + PCA/GLM + PCA) in Experiment 1.

Features	Classifier	Accuracy	Precision	Recall	F1-Score	AUC
CNN combination vector (Lasso+PCA/GLM+ PCA)	SVM	0.614/0.600	0.608/0.582	0.614/0.600	0.604/0.586	0.852/0.856
	MLP	0.807/0.821	0.815/0.828	0.807/0.821	0.803/0.817	0.939/0.951
	RF	0.636/0.657	0.634/0.653	0.636/0.657	0.621/0.653	0.878/0.872
	LR	0.700/0.664	0.699/0.653	0.700/0.664	0.685/0.648	0.889/0.861
	GB	0.657/0.636	0.664/0.635	0.657/0.636	0.658/0.629	0.889/0.863
	LDA	0.664/0.650	0.675/0.656	0.664/0.650	0.661/0.648	0.902/0.882

**Table 7 diagnostics-12-02274-t007:** The different ML classifiers’ performance on the test set of lung radiomics combination vector (Lasso + PCA/GLM + PCA) in Experiment 1.

Features	Classifier	Accuracy	Precision	Recall	F1-Score	AUC
Lung radiomics combination vector (Lasso+PCA/GLM+ PCA)	SVM	0.615/0.531	0.617/0.524	0.615/0.531	0.615/0.525	0.871/0.798
	MLP	0.797/0.811	0.803/0.815	0.797/0.811	0.795/0.809	0.941/0.956
	RF	0.713/0.699	0.719/0.706	0.713/0.699	0.710/0.701	0.913/0.904
	LR	0.727/0.657	0.724/0.650	0.727/0.657	0.725/0.649	0.899/0.873
	GB	0.762/0.713	0.762/0.712	0.762/0.713	0.761/0.710	0.931/0.912
	LDA	0.755/0.713	0.762/0.717	0.755/0.713	0.758/0.713	0.933/0.920

**Table 8 diagnostics-12-02274-t008:** The AMGNN classifier’s performance on the test set in Experiment 2.

Features	Classifier	Accuracy	Precision	Recall	F1-Score	AUC
3D CNN feature (13,824) (K = 2–6 (PCA) + d = 13,824)	AMGNN (K = 2)	0.793	0.789	0.793	0.774	0.933
	AMGNN (K = 3)	0.786	0.803	0.786	0.780	0.884
	AMGNN (K = 4)	0.807	0.831	0.807	0.799	0.929
	AMGNN (K = 5)	0.807	0.811	0.807	0.804	0.924
	AMGNN (K = 6)	0.800	0.825	0.800	0.795	0.927
	Mean	0.799	0.812	0.799	0.790	0.919
3D CNN feature (13,824) (K = 2–6 (GLM) + d = 13,824)	AMGNN (K = 2)	0.779	0.798	0.779	0.762	0.917
	AMGNN (K = 3)	0.821	0.824	0.821	0.811	0.929
	AMGNN (K = 4)	0.786	0.787	0.786	0.776	0.913
	AMGNN (K = 5)	0.807	0.819	0.807	0.808	0.941
	AMGNN (K = 6)	0.836	0.840	0.836	0.831	0.950
	Mean	0.806	0.814	0.806	0.798	0.930
Lung radiomics feature (1316) (K = 2–6 (PCA) + d = 1316)	AMGNN (K = 2)	0.864	0.864	0.864	0.862	0.966
	AMGNN (K = 3)	0.879	0.880	0.879	0.878	0.972
	AMGNN (K = 4)	0.850	0.851	0.850	0.850	0.952
	AMGNN (K = 5)	0.900	0.899	0.900	0.900	0.972
	AMGNN (K = 6)	0.871	0.872	0.871	0.865	0.948
	Mean	0.873	0.873	0.873	0.871	0.962
Lung radiomics feature (1316) (K = 2–6 (GLM) + d = 1316)	AMGNN (K = 2)	0.864	0.880	0.864	0.861	0.948
	AMGNN (K = 3)	0.857	0.858	0.857	0.857	0.971
	AMGNN (K = 4)	0.864	0.869	0.864	0.862	0.943
	AMGNN (K = 5)	0.871	0.871	0.871	0.870	0.964
	AMGNN (K = 6)	0.864	0.864	0.864	0.862	0.973
	Mean	0.864	0.868	0.864	0.862	0.960

**Table 9 diagnostics-12-02274-t009:** The AMGNN classifier’s performance on the test set in Experiment 3.

Features	Classifier	Accuracy	Precision	Recall	F1-Score	AUC
Selected 3D CNN feature (60) (K = 2–6 (PCA) + d = 60)	AMGNN (K = 2)	0.800	0.816	0.800	0.801	0.927
	AMGNN (K = 3)	0.814	0.812	0.814	0.809	0.947
	AMGNN (K = 4)	0.836	0.846	0.836	0.833	0.957
	AMGNN (K = 5)	0.829	0.829	0.829	0.827	0.962
	AMGNN (K = 6)	0.807	0.818	0.807	0.809	0.943
	Mean	0.817	0.824	0.817	0.816	0.947
Selected 3D CNN feature (60) (K = 2–6 (GLM) + d = 60)	AMGNN (K = 2)	0.800	0.797	0.800	0.797	0.941
	AMGNN (K = 3)	0.800	0.806	0.800	0.802	0.940
	AMGNN (K = 4)	0.814	0.837	0.814	0.812	0.949
	AMGNN (K = 5)	0.807	0.822	0.807	0.810	0.944
	AMGNN (K = 6)	0.793	0.798	0.793	0.793	0.939
	Mean	0.803	0.812	0.803	0.803	0.943
Selected lung radiomics feature (106) (K = 2–6 (PCA) + d = 106)	AMGNN (K = 2)	0.900	0.910	0.900	0.900	0.981
	AMGNN (K = 3)	0.907	0.914	0.907	0.908	0.983
	AMGNN (K = 4)	0.879	0.884	0.879	0.879	0.963
	AMGNN (K = 5)	0.879	0.879	0.879	0.878	0.954
	AMGNN (K = 6)	0.871	0.872	0.871	0.868	0.962
	Mean	0.887	0.892	0.887	0.887	0.969
Selected lung radiomics feature (106) (K = 2–6 (GLM) + d = 106)	AMGNN (K = 2)	0.879	0.889	0.879	0.879	0.951
	AMGNN (K = 3)	0.886	0.887	0.886	0.886	0.956
	AMGNN (K = 4)	0.871	0.882	0.871	0.875	0.969
	AMGNN (K = 5)	0.879	0.881	0.879	0.878	0.971
	AMGNN (K = 6)	0.886	0.886	0.886	0.881	0.963
	Mean	0.880	0.885	0.880	0.880	0.962

**Table 10 diagnostics-12-02274-t010:** The AMGNN classifier’s performance on the test set in Experiment 4.

Features	Classifier	Accuracy	Precision	Recall	F1-Score	AUC
3D CNN feature (13,824) (K = 2–6 (PCA) + d = 60)	AMGNN (K = 2)	0.793	0.793	0.793	0.791	0.933
	AMGNN (K = 3)	0.843	0.851	0.843	0.841	0.958
	AMGNN (K = 4)	0.800	0.801	0.800	0.798	0.941
	AMGNN (K = 5)	0.836	0.843	0.836	0.835	0.963
	AMGNN (K = 6)	0.793	0.798	0.793	0.794	0.847
	Mean	0.813	0.817	0.813	0.812	0.928
3D CNN feature (13,824) (K = 2–6 (GLM) + d = 60)	AMGNN (K = 2)	0.800	0.797	0.800	0.797	0.941
	AMGNN (K = 3)	0.800	0.806	0.800	0.802	0.940
	AMGNN (K = 4)	0.807	0.821	0.807	0.807	0.955
	AMGNN (K = 5)	0.814	0.832	0.814	0.819	0.958
	AMGNN (K = 6)	0.821	0.833	0.821	0.818	0.945
	Mean	0.808	0.818	0.808	0.809	0.948
Lung radiomics feature (1316) (K = 2–6 (PCA) + d = 106)	AMGNN (K = 2)	0.929	0.929	0.929	0.928	0.984
	AMGNN (K = 3)	0.893	0.912	0.893	0.895	0.979
	AMGNN (K = 4)	0.893	0.894	0.893	0.892	0.979
	AMGNN (K = 5)	0.871	0.886	0.871	0.876	0.971
	AMGNN (K = 6)	0.893	0.908	0.893	0.889	0.984
	Mean	0.896	0.906	0.896	0.896	0.979
Lung radiomics feature (1316) (K = 2–6 (GLM) + d = 106)	AMGNN (K = 2)	0.886	0.885	0.886	0.884	0.984
	AMGNN (K = 3)	0.943	0.946	0.943	0.943	0.984
	AMGNN (K = 4)	0.871	0.889	0.871	0.874	0.947
	AMGNN (K = 5)	0.879	0.886	0.879	0.879	0.979
	AMGNN (K = 6)	0.893	0.891	0.893	0.891	0.969
	Mean	0.894	0.899	0.894	0.894	0.973

**Table 11 diagnostics-12-02274-t011:** Comparison of the results of our proposed method with other previous methods.

Reference	Method	Feature	Accuracy	Precision	Recall (Sensitivity)	F1-Score	AUC	Specificity
Yang, Yingjian, et al. [25]	Lasso + MLP	CT-Based Radiomics	0.830	0.830	0.830	0.820	0.950	-
Spina, Gabriele, et al. [49]	Text representation + LDA	Multimodal Sleep Data	-	-	0.78	-	-	-
V K BAIRAGI, et al. [50]	CWT	Electromyography	0.859	0.849	0.882	-	0.865	0.855
Li, Zongli, et al. [26]	Variance threshold + Select K Best + Lasso + SVM	CT-Based Radiomics	0.759	0.834	0.723	0.771	0.799	0.805
Li, Zongli, et al. [26]	Variance threshold + Select K Best + Lasso + LR	CT-Based Radiomics	0.763	0.820	0.758	0.778	0.797	0.766
Our method	GLM + Lasso + AMGNN	CT-Based Radiomics	0.943	0.946	0.943	0.943	0.984	0.982

## Data Availability

The data supporting this study’s findings are available from the corresponding author upon reasonable request.

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
