# Peer review of "Lung Radiomics Features Selection for COPD Stage Classification Based on Auto-Metric Graph Neural Network"

_diagnostics, 2022, doi:10.3390/diagnostics12102274_

Round 1
Reviewer 1 Report
In this study, the authors proposed a COPD stage classification based on auto-metric graph neural network. The experimental results showed that the proposed approach outperforms other approaches. It is an interesting topic for readers. However, there are some problems listed as follows:
1. The last paragraph of section 1 showed the experimental results and I suggest that the authors need to move it to the experimental results. In this paragraph, the characteristics of the proposed approaches should be improved. The characteristics of the proposed Radiomics features extraction, generalized linear model, Lasso analysis, and AMGNN should be described in this paragraph.
2. In subsection 2.1, the number of subjects is limited and I suggest that the author can use N-fold cross-validation to objectively examine the proposed approaches.
3. For subsection 2.2.4, I suggest that the authors can move this subsection to section 3 “Results.”
4. For section 3, I suggest that the authors need to reorganize the experiments. Many factors of the proposed approaches should be systematically examined. For example, in subsection 3.1, the features of the proposed approach and 3D CNN were compared. We can find that the performance of 3D CNN is better that that of lung radiomics features (In table 2). When reducing the number of features, the selected lung radiomics feature outperforms the selected 3D CNN features. It is not a good experimental design, since the number of features for 3D CNN and lung radiomics is different. The authors can show the performance in a different number of features for 3D CNN and lung radiomics. Therefore, the reader can easily understand the effects of the number of features.
5. Why did the authors select 106 features by using the lasso algorithm? The authors need to examine the effects of the number of features.
6. To reduce the number of features, PCA, GLM, and lasso were examined in this study. However, the readers would like the see the effects of combining all or part of these methods.
7. The parameters of neural networks used in the experiments should be detailed.
Reviewer 2 Report
Overall, the manuscript shows an interesting approach to COPD stage classification using radiomics.
There are major spelling and grammar issues; among these:
- line 72: there is no subject in this sentence.
- line 78: change "mall" to "small".
- line 82: please rephrase the sentence: "reference [15] points out...", as it is not clear.
- line 97-99: rephrase as it is not clear.
I suggest revision by a native english speaker.
Figures are good. I suggest moving figure description from the text to the figure legend (as in Methods, line 170-172).
Discussion: please avoid repetitions; I suggest to remove line 421-424.
Reviewer 3 Report
This paper presents lung radiomics combination for COPD stage classification based on an autometric graph neural network with a meta-learning strategy. The authors some pre-processing techniques for images using ResU-Net. Lung radiomics features are extracted from the parenchyma images by PyRadiomics. The authors did some experiments to show accuracy, precision, and other metrics. Although the paper has merit, it needs further revision.
1. The title is very long and confusing. Please reduce the title in a meaningful way.
2. Why image segmentation is done using ResU-Net? Need justification.
3. The authors choose six common ML as classifiers. How they are selected? Why ML are chosen for image classification? Normally CNN models are used.
4. The authors should compare the results with other works. Not just with some ML techniques.
5. The authors should improve the reference sections with more recent works. Only few works have been referenced from 2020-2022 to justify the advancement of the work.
6. Please discuss the experiment environment and parameters setting for the proposed models’ training and testing. How the models handle biasness (overfitting and underfitting)
7. What about limitations
Round 2
Reviewer 1 Report
All issues I concerned have been revised carefully. I suggest this manuscript should be accepted as a journal paper.
Reviewer 3 Report
Thanks for addressing the comments. It is now in good shape.